# ASYMMETRIC PROXIMAL POLICY OPTIMIZATION: MINI-CRITICS BOOST LLM REASONING

**Jiashun Liu**[1,4*]  **Johan Obando-Ceron**[2,3*]  **Han Lu**[4]  **Yancheng He**[4]  **Weixun Wang**[4†]
**Wenbo Su**[4]  **Bo Zheng**[4]  **Pablo Samuel Castro**[2,3]  **Aaron Courville**[2,3]  **Ling Pan**[1†]
[1]HKUST   [2]Mila – Québec AI Institute   [3] Université de Montréal   [4]Alibaba Group

## ABSTRACT

Most recent RL for LLMs methods avoid explicit critics, replacing them with average advantage baselines. This shift is largely pragmatic: conventional value functions are computationally expensive to train at LLM scale and often fail under sparse rewards and long reasoning horizons. We revisit this bottleneck from an architectural perspective and introduce Asymmetric Proximal Policy Optimization (AsyPPO), a simple and scalable framework that restores the critic's role while remaining efficient in large-model settings. AsyPPO employs a set of lightweight *mini-critics*, each trained on disjoint prompt shards. This design encourages diversity while preserving calibration, reducing value-estimation bias. Beyond robust estimation, AsyPPO leverages inter-critic uncertainty to refine the policy update: (i) masking advantages in states where critics agree and gradients add little learning signal, and (ii) filtering high-divergence states from entropy regularization, suppressing spurious exploration. After training on open-source data with only 5k samples, AsyPPO consistently improves learning stability and performance across multiple benchmarks over strong baselines, e.g., GRPO, achieving performance gains of $> 6\%$ on Qwen3-4b-Base and about $3\%$ on Qwen3-8b-Base and Qwen3-14b-Base over classic PPO, without additional tricks. Such results underscore the value of architectural innovations for scalable, efficient learning.

## 1 INTRODUCTION

Proximal Policy Optimization (PPO) (Schulman et al., 2017) stands as one of the most powerful actor-critic algorithms in deep RL, and has demonstrated its potential across diverse domains such as computer games (Yu et al., 2022; Schwarzer et al., 2023) and robotics control (Raj & Kos, 2024). In the realm of large-scale language models (LLMs), PPO has also proven transformative and has been widely applied in the post-training stage to stimulate the reasoning ability of LLMs (Hu et al., 2025). However, the transition from classical RL to *RL for LLMs* (RL4LLM) introduces an unprecedented computational challenge, as LLMs operate at scales orders of magnitude larger than traditional RL environments. Directly applying PPO's default symmetric actor-critic design, where the critic is as large as the actor, creates significant computational overhead. In addition, training full critics at LLM scale is expensive and inaccurate under sparse, long-horizon rewards (Yuan et al., 2025b).

Faced with these challenges, the RL4LLM community has largely sidelined a key element of classical PPO – its critic. GRPO (He et al., 2025), and its variants, including GSPO (Zheng et al., 2025) in the Qwen series and DAPO (Yu et al., 2025), have achieved great success in replacing value functions with group sampling and average-advantage baselines for coarse-grained estimation of advantages. While effective, this paradigmatic shift abandons a key concept of RL: robust state value estimation can naturally mitigate training collapse caused by advantage bias (Wang et al., 2025b; Liu et al., 2024), especially under off-policy settings. This landscape motivates a fundamental reconsideration of architectural assumptions inherited from deep RL, prompting the following central question: **Can we achieve lightweight yet robust value estimation by redesigning PPO to depart from the standard symmetric actor–critic architecture, enabling stable and efficient learning?**

---

*Equal contribution: {ljshasdream, jobando0730}@gmail.com
†Corresponding authors: weixun.wwx@taobao.com, lingpan@ust.hk

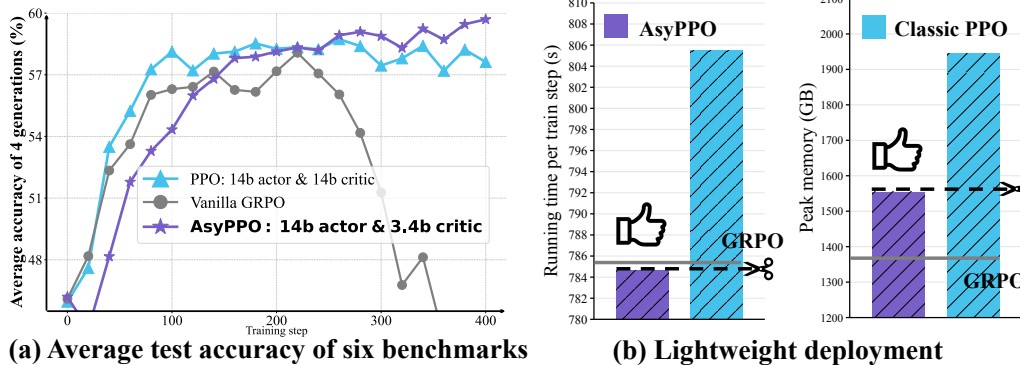

**(a) Average test accuracy of six benchmarks**     **(b) Lightweight deployment**

Figure 1: **(Left)**: **Learnable critics naturally enhance policy stability through fine-grained value estimation and yield continuous gains as training progresses.** Off-policy ratio=8, average@4 of 6 benchmarks, i.e., AIME 24, AIME 25, MATH-500, OlympiadBench, MinervaMath, and AMC 2023. **(Right)**: `AsyPPO` restores the critic's role in PPO while remaining lightweight and stable under LLM-scale training. The average clock time of training and the peak GPU memory usage of `AsyPPO` are significantly lower than those of the classic PPO, remain at the **GRPO level**.

To fill this research gap, we begin with a key insight: the initial rich representational ability inherited from pre-trained models significantly enhances the feasibility of the asymmetric actor-critic in the RL4LLM domain, unlike agents that learn from scratch in classical deep RL, painting a promising prospect for lightweight deployment and computational efficiency. Our initial experiments validate this hypothesis, where we find that a small critic, e.g., `Qwen3-0.6b-Base`, can indeed provide meaningful guidance to a much larger actor, e.g., `Qwen3-8b-Base`, demonstrating meaningful performance improvements over the base model. However, this asymmetric setup underperforms classical symmetric PPO, revealing limitations in a single small critic's value estimation capabilities.

To unlock the capabilities of the small critic, we consider critic ensembles to improve its value estimation and policy guidance. However, naive ensembles offer limited benefits for policy learning, as LLM critics start from identical pre-trained checkpoints with different heads only and are trained on the same data, leading to nearly identical behaviors that provide no corrective benefit. To tackle this critical challenge, we propose a simple yet effective non-overlapping data partitioning technique, in which each critic is trained via the subset formed by uniformly extracting responses from each prompt without overlap. This design encourages diversity among small critics and mitigates the risk of perception asynchrony among critics. Leveraging our ensemble-based value correction, small critics can provide reliable guidance to large policies despite their limited expressivity (Tint et al., 2024). Surprisingly, we find that double critic could be the sweet spot between correction capability and efficiency, it yields a qualitative leap in evaluation reliability while incurring the minimal redundancy needed for bias reduction. More critics increase the computation without proportional gains. Empirically, we demonstrate that two `Qwen3-1.7b-Base` critics robustly guide a larger policy, e.g., `Qwen3-14b-Base`, reducing critic over-parameterization while outperforming symmetric PPO under off-policy setting (Figure 1(a)). Notably, asymmetric architecture reduces peak memory by 20%, and accelerates training by around 20 seconds per step (see Figure 1(b)).

We further discovered that the agreement and divergence patterns in value estimates between our double critics, measured by their standard deviation, provide a useful signal for refining the policy loss objective. Value-estimation heterogeneity reflects both uncertainty and informativeness of the states. Leveraging this, we mask advantage values in states where critics strongly agree, reducing overfitting to low-quality samples and improving training stability. Conversely, we exploit divergence across critics by filtering out uncertain states from entropy regularization, since such states often correspond to low-probability continuations or spurious, reasoning-irrelevant patterns that inject noise into entropy measurements (Ahmed et al., 2019). Thus, restricting entropy regularization to high-confidence states promotes safer exploration and improves performance. With this refined loss objective, the policy further amplifies its learning efficiency ( Figure 2). Overall, we refer to the above components as Asymmetric Proximal Policy Optimization (`AsyPPO`). The contributions of `AsyPPO` can be summarized in three main aspects:

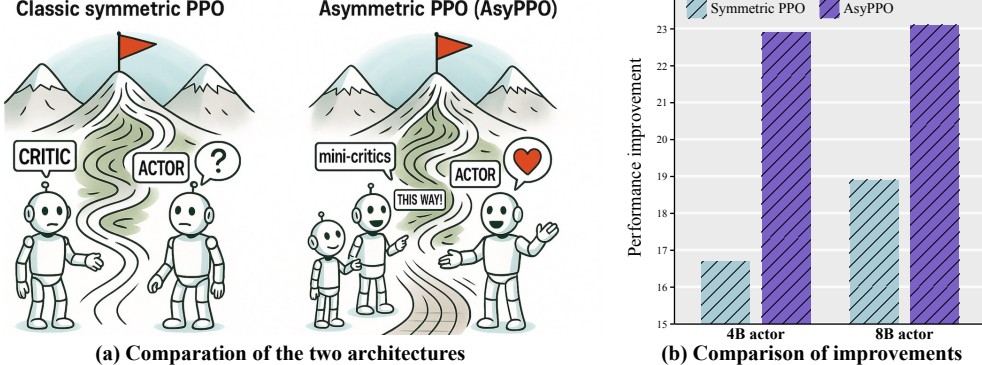

**(a) Comparation of the two architectures**      **(b) Comparison of improvements**

Figure 2: **Visual intuition behind `AsyPPO`.** Small yet expressive critics can guide larger actors effectively when their representations are well-initialized. **(a)**: A single critic struggles to align its value signal, leading to uncertain policy updates. Two small critics reach consensus ("This way") and provide robust, low-variance guidance to the actor. **(b)**: By leveraging representational priors and critic agreement, `AsyPPO` significantly outperforms classic PPO in terms of performance on various policy models, i.e., `Qwen3-4b-Base` and `Qwen3-8b-Base`. The Y-axis represents the improvement for the initial policy. The score calculation is the same as that in Figure 1.

---

1. **Robust Estimation:** Prompt-level data partitioning enhances ensemble reliability and yields consistent performance improvements. (§4.1)

2. **Lightweight Architecture:** The asymmetric design mitigates critic over-parameterization and opens a new direction for RL4LLM. (§4.1)

3. **Objective Refinement:** We introduce two uncertainty-aware modifications to the PPO objective that improve sample efficiency and enable safer exploration. (§4.2)

---

## 2   PRELIMINARIES

**Proximal Policy Optimization (PPO).** PPO (Schulman et al., 2017) is a widely used actor-critic algorithm in the policy gradient family. It improves the stability by optimizing a *clipped surrogate objective*, which limits how much the updated policy $\pi_\theta$ can deviate from the old policy $\pi_{\theta_{\text{old}}}$ at each update step. The objective is defined as:

$$\mathcal{J}_{\text{PPO}}(\theta) = \mathbb{E}_{\left[q \sim P(Q),\ o \sim \pi_{\theta_{\text{old}}}(O|q)\right]}$$
$$\frac{1}{|o|} \sum_{t=1}^{|o|} \min\left( \frac{\pi_\theta(o_t|q, o_{<t})}{\pi_{\theta_{\text{old}}}(o_t|q, o_{<t})} A_t,\ \text{clip}\left( \frac{\pi_\theta(o_t|q, o_{<t})}{\pi_{\theta_{\text{old}}}(o_t|q, o_{<t})},\ 1-\epsilon,\ 1+\epsilon \right) A_t \right), \quad (1)$$

where $\pi_\theta$ and $\pi_{\theta_{\text{old}}}$ denote the current and previous policy, respectively. Here $q$ is a sampled *question* and $o$ the generated *output sequence*,, with $o_t$ the $t$-th token. $\epsilon$ is the clipping hyperparameter that constrains the update ratio. $A_t$ is the advantage estimate at step $t$, typically computed with Generalized Advantage Estimation (GAE) (Schulman et al., 2015).

**Generalized Advantage Estimation (GAE).** GAE addresses the bias–variance trade-off in advantage estimation by combining multi-step returns with exponentially decaying weights:

$$\hat{A}_t^{\text{GAE}(\gamma,\lambda)} = \sum_{l=0}^{\infty} (\gamma\lambda)^l \delta_{t+l}, \qquad \delta_t = r_t + \gamma V(s_{t+1}) - V(s_t).$$

Here $V(s)$ is the value function, $\gamma \in [0, 1]$ is the discount factor, and $\lambda \in [0, 1]$ is the GAE parameter that balances bias and variance. Setting $\lambda = 0$ recovers the low-variance, high-bias $TD(0)$ estimator, while $\lambda = 1$ corresponds to the high-variance, low-bias Monte Carlo estimator. In practice, PPO

leverages GAE together with the clipped objective, yielding stable training and improved sample efficiency. The choice of $\gamma$ and $\lambda$ critically influences the temporal horizon and smoothness of the advantage estimates, and thus the convergence of the policy.

# 3 RELATED WORK

**Critic-based RL4LLM algorithms** Shao et al. (2024) first demonstrated that large-scale reinforcement learning (RL) with outcome-based rewards can unlock long-tail reasoning, beginning from an unaligned base model. This finding has led to numerous variations of the Proximal Policy Optimization (PPO) algorithm. As far as we know, most algorithm research is mainly based on the baseline normalized advantage calculation method (Hu, 2025; Liu et al., 2025c; Chen et al., 2025a). On the other hand, value-based algorithm innovations are relatively few, Yuan et al. (2025b) argued that the decay factor is not well-suited for complex reasoning tasks that require long chains of thought (CoT). Yue et al. (2025); Zhu et al. (2025); Zhao et al. (2025) proposed novel mechanisms to enhance the robustness of the critic model when faced with noisy reward signals. Open-Reasoner-Zero (Hu et al., 2025) argues that vanilla PPO without KL regularization suffices to scale training stably. T-PPO (Fan et al., 2025) uses critic to enhance the stability of policy training in the long-tail asynchronous setting (Fu et al., 2025). Another similar research line to introduce critic-like models is done with the introduction of Implicit PRM (Yuan et al., 2025a). This approach is also able to provide token-level supervision for scalable RL training. PRIME (Cui et al., 2025a) adapted a specific reward model formulation to directly generate token-level rewards. However, current mainstream RL4LLM algorithms primarily emphasize critic-free optimization (Zhang et al., 2025). In this context, our research aim to underscore the importance of the critic in RL4LLM scenarios and try to address the deployment limitations associated with critics.

**Asymmetric architecture.** In the realm of continuous deep RL, recent studies have investigated the potential of asymmetric network structures by reducing the capacity of the actor network. For example, Mastikhina et al. (2025); Mysore et al. (2021) suggest that the actor can function effectively with a significantly smaller capacity compared to the critic. Empirical evidence from Tan et al. (2023) supports this idea, demonstrating that sparsifying the policy network can enhance effective policy learning while significantly improving both inference and training speeds. Additionally, Liu et al. (2025a) found that pruning the actor network's topology based on trial gradients can yield better performance. Similarly, Ma et al. (2025) revealed that even random pruning of the actor network can maintain performance within the SimBa network architecture (Lee et al., 2025). These contributions highlight the adaptability of RL in accommodating asymmetric designs, providing valuable insights for our research. However, existing works primarily concentrate on reducing the actor's size within simple network frameworks. Our paper pioneers the exploration of effectively guiding small critics to inform larger actors by optimizing PPO within the RL4LLM scenario.

# 4 ASYMMETRIC PROXIMAL POLICY OPTIMIZATION

We begin by empirically examining the potential of the asymmetric actor-critic framework while highlighting the limitations of naive ensemble critics in LLM reasoning. By analyzing key differences between classical deep RL and RL4LLM, we propose a group-level non-overlapping data division strategy that enables lightweight mini-critics to provide reliable value estimation (§4.1). Building on this, we investigate the role of divergence and agreement among the mini-critics and find that uncertainty in their value estimates carries strong representational power for measuring sample quality. Leveraging this insight, we incorporate value uncertainty as a signal into the policy optimization objective, reformulating the loss function and refining the entropy regularization to improve sample efficiency and exploration capability of the policy (§4.2).

## 4.1 TOWARDS LIGHTWEIGHT VALUE ESTIMATION

In LLM reasoning, the policy inherits expressive capabilities from the pre-trained model at initialization. As shown in Figure 3 (Left), even without critic warm-up, a small critic, i.e., Qwen3-0.6B-Base (Yang et al., 2025), can provide useful guidance, demonstrating the potential of an asymmetric architecture. However, due to sparse rewards and the small critic's limited

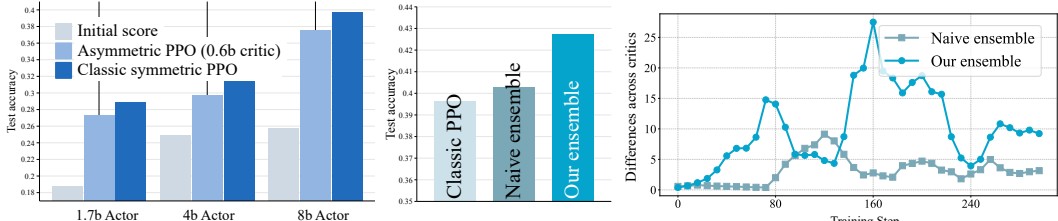

Figure 3: **Left:** The single mini-critic parameterized by `Qwen3-0.6b-Base` can effectively guide policies across model scales. **Middle:** There are significant differences in the guiding ability of the two ensemble critics for policies. Actors uniformly use `Qwen3-8B-Base`, while critics use `Qwen3-0.6B-Base`. **Right:** Our ensemble method intensifies the cognitive differences among mini-critics. The y-axis represents the standard deviation between the values calculated by the two mini-critics. We train on 5,000 questions sampled from DeepMath-103K (He et al., 2025) and evaluate policies on five challenging math benchmarks: AIME 2024, MATH-500, OlympiadBench, MinervaMath, and AMC 2023. For each question, we report the average of 4 generations.

familiarity with long-tail reasoning trajectories favored by larger models (Li et al., 2025), its value estimates are often inaccurate, leading to suboptimal policy guidance compared to symmetric PPO.

**Starting from the ensemble system.** To strengthen mini-critic perceptual capacity, we first adopt an ensemble of critics, a standard technique in classical deep RL for reducing estimation bias (Chen et al., 2021). In practice, we add a second critic based on the same base model and average their predictions for value estimation. These corrected values are then used in advantage computation via GAE. However, as Figure 3 (Middle) shows, this naive ensemble approach yields limited improvement. The reason becomes clear in Figure 3 (Right), the two mini-critics exhibit nearly identical behavior, failing to provide the diversity that ensembles rely on. In classical RL, critics are initialized randomly, ensuring parameter diversity and differentiated value estimates, which is essential for ensemble effectiveness. By contrast, in RL4LLM, critics are typically initialized from the same pretrained model, which accelerates learning but reduces diversity. This motivates a question: *under homogeneous initialization, can ensemble critics remain effective in LLM reasoning?*

**Group level non-overlap data division.** Beyond explicitly increasing parameter differences through initialization, another promising approach is to provide differentiated optimization signals for each critic during training. Intuitively, training critics on non-overlapping subsets of data encourages them to learn from distinct trajectories and reward distributions, steering their updates in different directions and promoting functional diversity. However, in practice, randomly partitioning the training data can lead to asynchronous perception at the prompt level, where critics encounter inconsistent reasoning patterns from different questions. This imbalance increases the risk of overfitting to specific response types, resulting in unstable discrepancies in value estimates. In extreme

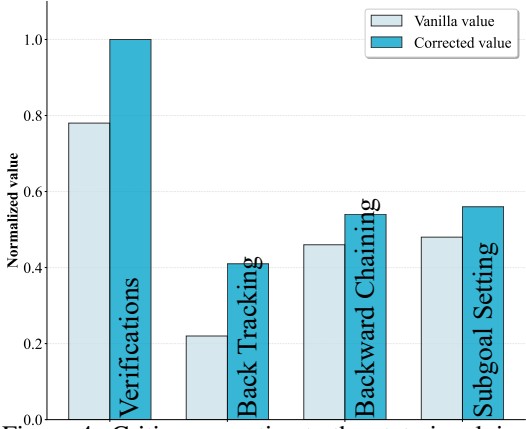

Figure 4: Critics can estimate the state involving key patterns, according to Gandhi et al. (2025).

cases, such divergence may cause policy collapse. To mitigate this, we uniformly divide the data into disjoint subsets at the prompt level, ensuring that each critic receives an equal share of responses within every prompt (or group). This design maintains perceptual synchrony across critics within each question while creating differentiated rewards and observations. Our ensemble critic training process can then be formalized as:

$$\mathcal{L}_{\text{critic}}(\phi) = \sum_{m=1}^{M} \mathcal{L}_{\text{critic}}^{(m)}(\phi_m) = \sum_{m=1}^{M} \mathbb{E}_{(s_t, R_t) \sim \mathcal{D}_m} \left[ (V(s_t; \phi_m) - R_t)^2 \right], \tag{2}$$

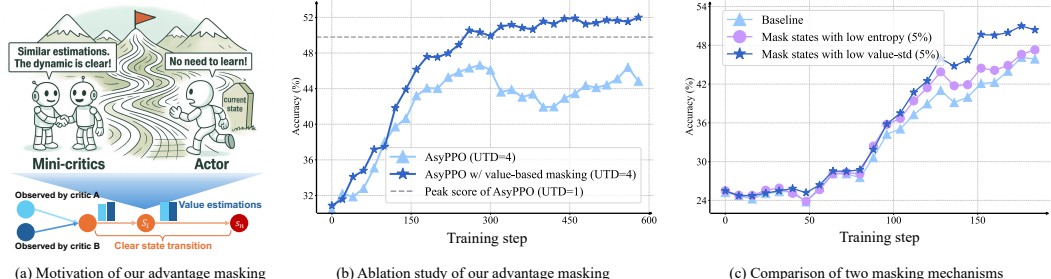

(a) Motivation of our advantage masking  (b) Ablation study of our advantage masking  (c) Comparison of two masking mechanisms

Figure 5: **(a):** Agreement among critics implies the state's downstream dynamics are well modeled by the policy, making these samples low-value for learning and best avoided for overfitting. **(b):** In the high data-reuse setting (UTD=4), masking the bottom $20\%$ (by value-std) boosts AsyPPO's learning efficiency, **yields an improvement of about 6 points**. The accuracy records of the six benchmarks follow Figure 1 (b). **(c):** We evaluated two $5\%$ masking mechanisms on vanilla AsyPPO (baseline), i.e., entropy vs. value-std. The value-std masking produced the strongest learning efficiency benefit. Actors use `Qwen3-4B-Base`, while critics use `Qwen3-0.6B-Base`.

$M$ denotes the number of mini-critic with parameters $\{\phi_m\}_{m=1}^{M}$. Each critic aim to fit the return $R_t$ based on its assigned subset $\mathcal{D} = \bigcup_{m=1}^{M} \mathcal{D}_m, \mathcal{D}_i \cap \mathcal{D}_j = \emptyset$. Corrected advantage $\bar{A}$ can be obtained:

$$\bar{A}_t(\gamma, \lambda) = \sum_{l=0}^{T-t-1} (\gamma\lambda)^l \delta_{t+l}, \ \delta_t = r_t + \gamma\bar{V}(s_{t+1}) - \bar{V}(s_t); \ \bar{V}(s_t) = \frac{1}{M}\sum_{m=1}^{M} V_m(s_t; \phi_m) \quad (3)$$

The results in Figure 3 (Middle, Right) demonstrate that critics trained under our ensemble strategy exhibit clearly differentiated behaviors. Statistical analysis from a linguistic perspective (Figure 4) reveals that the corrected values from our ensemble framework significantly encourage the policy to acquire core reasoning patterns. Overall, our method effectively unlocks the efficiency of asymmetric PPO and points to a promising new direction for RL4LLM algorithm design.

> **Takeaway 1**
>
> Optimizing the ensemble critic design enhances the learning capacity of the asymmetric actor–critic while significantly reducing computational overhead.

## 4.2 POLICY LOSS RECONSTRUCTION

Beyond enabling robust value estimation, we conjecture that ensemble mini-critics can further enhance policy learning efficiency. Intuitively, the degree of agreement among critics' value estimates for a given state can serve as a meaningful signal for policy optimization. This insight arises from our analysis of value fitting dynamics (Lee et al., 2021): when critics produce similar value estimates for a state $s_i$, it often indicates that $s_i$ is **low-informative**. Such states are frequently encountered across trajectories, and the rewards they yield exhibit low variance, causing critics to converge in their predictions, as visualized in Figure 5 (a). Analysis in Appendix B shows the positive correlation between value-std and the policy gradient, supporting the above speculation.

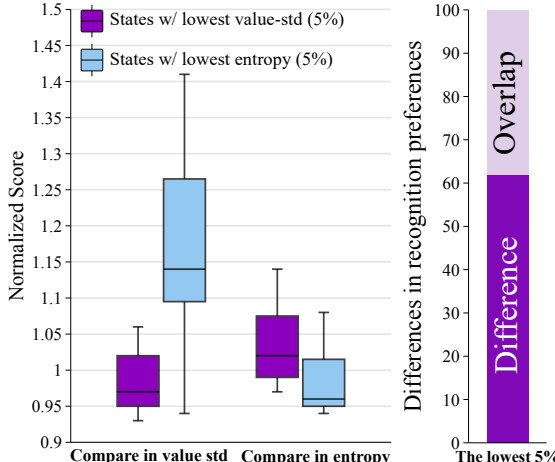

Figure 6: **Left:** States with low value-std maintain low entropy, but states with low entropy may have high value-std. **Right:** States with low entropy and states with low value-std show differences.

**Advantage masking based on the value agreement.** Recent studies show that preventing the policy from overfitting to low-information samples can substantially improve learning efficiency (Liu et al., 2025b). Since the degree of agreement across critics reflects state informativeness, where high agreement implies low uncertainty and limited learning potential, we use the standard deviation of critics' outputs to quantify the benefit of optimizing a given state. Specifically, we identify the top $k$ percentage of states with the highest agreement (i.e., lowest standard deviation) and mask their corresponding advantages in the policy loss. This suppresses gradient updates from low-informative transitions, filtering out noisy or redundant learning signals directing policy optimization toward higher-value data. The resulting policy loss objective is:

$$\mathcal{J}_{\text{PPO}}(\theta) = \mathbb{E}\frac{1}{|o|}\sum_{t=1}^{|o|}\mathbb{I}^A \cdot \min\left(\mathcal{IS}_t \cdot \bar{A}_t,\, \text{clip}\left(\mathcal{IS}_t,\, 1{-}\epsilon,\, 1{+}\epsilon\right)\bar{A}_t\right);\ \mathbb{I}_t^A = \begin{cases} 0, & \text{if } \sigma_t \in Low_k(\sigma) \\ 1, & \text{otherwise} \end{cases}$$

(4)

Here, $\sigma_t = std\left(\{V(s_t; \phi_m)\}_{m=1}^M\right)$ denotes the agreement of value estimates for state $s_t$. Important sampling is defined as $\mathcal{IS}_t = \frac{\pi_\theta(a_t|s_t)}{\pi_{\theta_{\text{old}}}(a_t|s_t)}$. Figure 5(b) shows that, masking the advantages corresponding to the 20% of states with high critic convergence, the policy exhibits stable learning dynamics even under high sample reuse (update-to-data ratio (UTD) = 4, i.e., each sample was used for training four times) and significantly improves sample efficiency. We further compared value-std (critic-side uncertainty) with entropy (policy-side uncertainty) (Wang et al., 2025a; Rahn et al., 2024; Cui et al., 2025b) by masking an equal fraction of states per step according to each metric. Figure 5(c) shows that value-std–based masking consistently delivers stronger learning benefits. This observation echoes classic RL findings (Osband et al., 2016), where ensemble-based value uncertainty acts as a proxy for learning dynamics. Figure 6 reveals that low value-std states consistently align with low entropy, suggesting that value-std is a precise uncertainty metric.

> **Takeaway 2**
>
> Agreement among critics provides a reliable measure of the learning benefit of the states.

**Entropy filtering based on value divergence.** When critics exhibit significant divergence in their evaluation of a state $s_j$, reflected in a high standard deviation, it may indicate that $s_j$ is *reasoning-independent*. For instance, different critics may encounter divergent reward distributions for trajectories passing through $s_j$, due to factors such as inference-irrelevant tokens or inherent semantic patterns in model generations. With a large $\lambda$, the dispersion in returns distribution propagates back to each state, amplifying disagreement among critics. In such cases, persistent exploration at $s_j$ is meaningless, as it does not correspond to an actionable decision state (Figure 7 (a)). To promote meaningful exploration while avoiding wasteful updates on noisy or non-decision states, we introduce a safe entropy regularization weighted by $\beta$. Specifically, we filter out states with high value std deviation when computing entropy $\mathcal{H}$. Complete policy loss is rewritten as:

$$\mathcal{J}_{\text{PPO}}(\theta) = \mathbb{E}_{\left[q \sim P(Q),\ o \sim \pi_{\theta_{\text{old}}}(O|q)\right]}\frac{1}{|o|}\sum_{t=1}^{|o|}\left[\mathbb{I}_t^A \cdot \min\left(\mathcal{IS}_t \cdot \bar{A}_t,\, \text{clip}\left(\mathcal{IS}_t,\, 1{-}\epsilon,\, 1{+}\epsilon\right)\bar{A}_t\right)\right.$$

$$\left. + \beta \cdot \mathbb{I}_t^{\mathcal{H}} \cdot \mathcal{H}\left[\pi_\theta(\cdot|s_t)\right]\right];\ \mathbb{I}_t^{\mathcal{H}} = \begin{cases} 0, & \text{if } \sigma_t \in top_h(\sigma) \\ 1, & \text{otherwise} \end{cases}.$$

(5)

Figure 7 (b) shows that, unlike naive entropy loss, which can yield suboptimal learning, our entropy regularization mitigates entropy collapse and stabilizes policy learning, avoiding spurious exploration while guiding the policy toward better convergence with higher returns. We also compare filtering based on value-std versus entropy. As shown in Figure 7 (c), the overlap between the two sets is minimal. Even after filtering the top 40% of hight value-std, policy entropy remains stable, while filtering the same fraction of high-entropy states causes entropy collapse. Statistical analysis of filtered tokens (Appendix C) confirms that removed words are typically adverbs, interjections that are irrelevant to decision-making. Algorithm 1 shows the pipeline of `AsyPPO`.

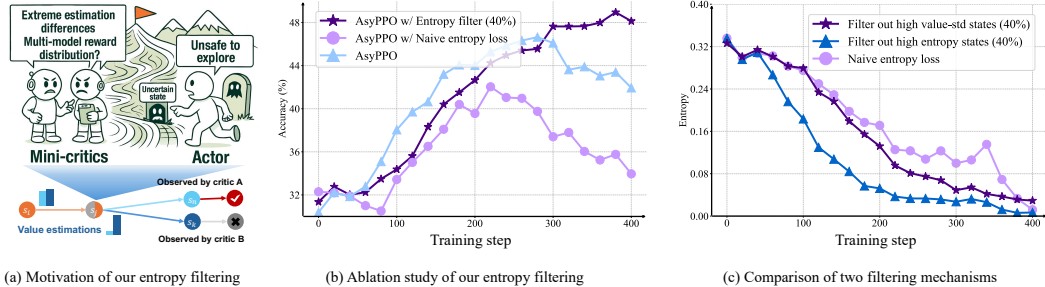

(a) Motivation of our entropy filtering   (b) Ablation study of our entropy filtering   (c) Comparison of two filtering mechanisms

Figure 7: **(a):** When critics diverge, the state is weakly coupled to the final outcome and has complex future dynamics; exploration in such non-critical states should be avoided. **(b):** Excluding states with high value-estimate standard deviation from the entropy loss prevents policy collapse induced by naive entropy regularization and **yields a roughly 7 percentage-point improvement**. The setup follows the settings in Figure 1 (b). **(c):** Excluding the top $40\%$ of high value-std states from the entropy loss preserves policy entropy at levels comparable to naive entropy guidance, whereas filtering the same percentage of states with the highest entropy collapse. The settings are consistent with Figure 5.

> **Takeaway 3**
>
> Divergence among value estimations indicates the cost-effectiveness of exploring the states.

---

**Algorithm 1:** Asymmetric PPO with two mini-critic

---

1   $\pi_\theta$: actor. $V_{\phi_{\{1,2\}}}$: mini-critics, $o_t \in O$: generation up to step $t$ in response $o$ under prompt $q$, $O$ denotes the total response in the batch. $\sigma(O)$: value estimation std across the critics. $\bar{A}$: corrected advantage. $\mathbb{I}^A$: The index for advantage masking. $\mathbb{I}^{\mathcal{H}}$: The index for entropy filtering.

2   **while** *training step < maximum step* **do**

3     $O \leftarrow \pi_\theta(Q)$

4     Build training subsets for each critic, and update $V_{\phi_{\{1,2\}}}$ according to Eq.2

5     $\bar{A} \leftarrow GAE(\bar{V}, r), \bar{V} \leftarrow mean(V_{\phi_1}(Q,O), V_{\phi_2}(Q,O))$ via Eq.3

6     Generate masking vector $\mathbb{I}^A \leftarrow Low_k(\sigma(O))$ and filtering vector $\mathbb{I}^{\mathcal{H}} \leftarrow Top_h(\sigma(O))$.

7     Update $\pi_\theta$ via reconstructed PPO loss (Eq.5)

---

## 5   EXPERIMENTS

In §4, we demonstrated the efficacy of `AsyPPO` on 4B and 8B LLMs (please refer to Figure 2(b) for results) through diversified experiments. This section examines `AsyPPO` more broadly through a suite of experiments. We organize the subsequent studies around three research questions: **RQ1:** Can `AsyPPO` and naive asymmetric PPO unlock general reasoning in larger LLM? **RQ2:** How sensitive is `AsyPPO` to the size and number of critics? **RQ3:** What setups are effective for advantage masking and entropy filtering?

### 5.1   GENERALIZATION TO LARGE MODELS

**Setup.** To ensure consistency with prior research, we fix the global batch size to 1024, with a maximum response length of 8192 tokens. The learning rate is set to $1e-6$. For text generation, we use a top_p value = 0.99, and top_k value = 100, temperature 0.99, $UTD = 4$ (also referred to as PPO_epoch, result in off-policy). The actor is `Qwen3-14b-Base`, while critics vary in size from the `Qwen3-Base` family. To ensure reproducibility and fairness, we exclusively use open-source datasets. We use the hard training dataset from Liu et al. (2026); Zeng et al. (2025), which exposes clear performance gaps across algorithms in long-tail reasoning tasks. We report the average@4

across 4 challenging benchmarks, i.e., MATH-500 (Lightman et al., 2023), OlympiadBench (He et al., 2024), MinervaMath (Lewkowycz et al., 2022), and AMC 2023 (?).

**Baselines.** For all algorithms, actors are initialized using `Qwen3-14b-Base`. Naive asymmetric PPO uses a single critic, i.e., `Qwen3-1.7b-Base`, `Qwen3-4b-Base` and `Qwen3-8b-Base`, and optimize with the vanilla PPO optimization objective. `AsyPPO` employs two mini-critics with advantage masking at 20% and entropy filtering at 20%. We use the setting of GRPO recommended by Liu et al. (2026). Full hyperparameter details are provided in Appendix A.3.

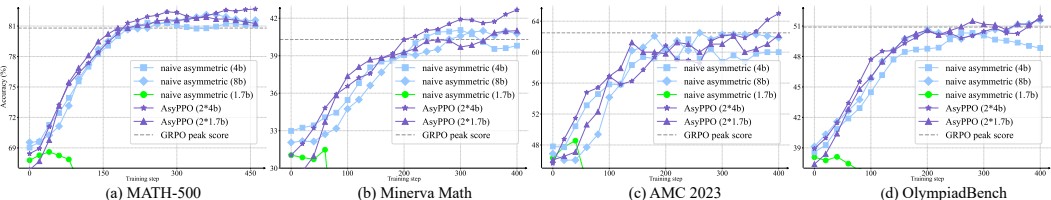

Figure 8: `AsyPPO` improves accuracy by **an average of about 3 + points** compared to GRPO, and achieves more than **20% lighter weight** than symmetrical PPO. Our naive asymmetric PPO still works on the 14b policy, but fails under the 1.7b critic setting. However, `AsyPPO` unlocks the 1.7b critic's ability to guide the 14b actor.

**Results.** Figure 8 shows that `AsyPPO` with two 4b critics achieves the strongest results across all tasks. Compared to GRPO, `AsyPPO` improves accuracy by an average of about 3 points. For naive asymmetric PPO (a single mini-critic guiding a large actor), we observe a clear critic-capacity threshold: single `Qwen3-1.7b-Base` critic cannot reliably guide 14b actors, despite successfully guiding an 8B actor; upgrading to a 4B critic restores effective learning. By contrast, `AsyPPO` lowers this requirement, 1.7b critics deliver substantial reasoning gains. Combined with the lightweight deployability in Figure 1(c), `AsyPPO` establishes an efficient and practical RL4LLM design.

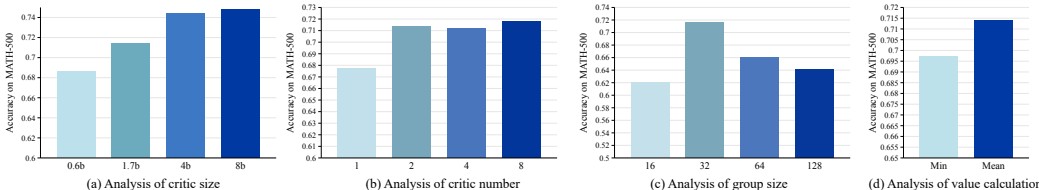

Figure 9: **(a):** The increase in the size of the critic further enhances the effectiveness of `AsyPPO`, which can be regarded as the marginal benefit brought by the parameter scaling up. We initialize the actor using `Qwen3-8b-Base` and initialize the double mini critic using four sizes of the Qwen3 Base model. **(b):** A qualitative improvement in performance can be achieved by using two mini critics. **(c):** A suitable group size for `AsyPPO` is 32. **(d):** Using the mean of the critic's estimated value can achieve better correction of the value than using min. For (b,c,d), we initialize the actor using `Qwen3-8b-Base` and initialize the mini critics using `Qwen3-1.7b-Base`.

## 5.2 ABLATION STUDY

The preceding results show that `AsyPPO` consistently enhances reasoning in LLMs across scales. We provide a module-wise analysis to characterize the algorithm from multiple perspectives.

**Ensemble critic system.** Figure 9 (a) shows a scaling-law–like trend: increasing critic size steadily raises the policy's peak score. We recommend using the largest critic model that fits in GPU memory to maximize `AsyPPO`'s optimization capacity. However, we do not see comparable gains from increasing the number of critics: Figure 9 (b) shows that two mini-critics are sufficient for a clear step-change in performance. Varying the GRPO group size (trajectories per prompt) while keeping other parameters at their defaults (Figure 9 (c)), and found 32 to be a robust setting. Comparing ensemble value aggregation (Figure 9 (d)), the mean of values outperforms the min value, suggesting overestimation is not a dominant issue in RL4LLM.

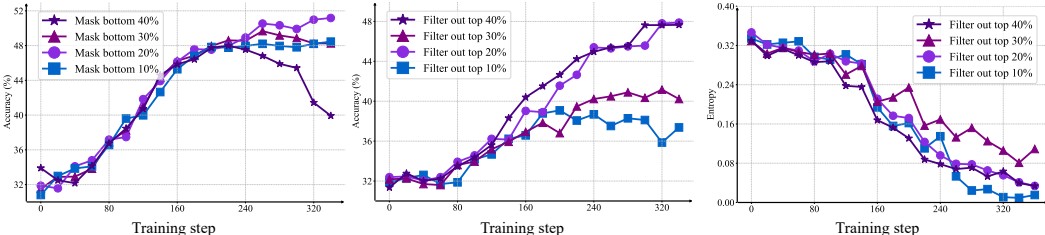

Figure 10: **Left**: Average test score on six benchmarks under various advantage masking setup. **Middle**: Average test score under various filtering out setup. **Right**: entropy curves during training. All experiments were based on `Qwen3-8b-Base` actor and `Qwen3-1.7b-Base` critic. The accuracy calculation follow Figure 3.

**Value-convergent-based advantage masking.** To identify a robust advantage-masking percentage, we adopt the main experiment settings with `Qwen3-8b-Base` as the policy and two `Qwen3-1.7b-Base` critics. Figure 10 (Left) shows that masking 20% of low-value-std states provides the strongest gains.

**Value-divergence-based entropy filter.** To find an appropriate filtering percentage, we follow the same setup as for advantage masking. We test masking 10%, 20%, 30%, and 40% of the highest-value-std states from the entropy loss. As shown in Figure 10 (Middle, Right), larger masks induce entropy collapse, while 20% strikes the best exploration–exploitation balance.

## 6  CONCLUSION

We reframed the critic bottleneck in RL4LLM as an architectural rather than a purely algorithmic or optimization issue. Our proposed Asymmetric Proximal Policy Optimization (`AsyPPO`) reinstates the critic's role via double lightweight mini-critics trained on disjoint prompt-level data, yielding diverse yet calibrated value estimates. Beyond improving value estimation robustly, we showed that inter-critic uncertainty provides an actionable signal for policy optimization: masking advantages for low-informativeness states and filtering high-divergence states from entropy regularization both reduce overfitting and promote safer, more effective exploration. Across standard LLM reasoning benchmarks, `AsyPPO` consistently improves general reasoning for models of varied sizes, empirically supporting asymmetric actor–critic design as a viable and efficient direction for RL4LLM. `AsyPPO` mitigates critic over-parameterization while improving sample and compute efficiency.

**Limitations** To ensure fairness and reliability under limited GPU resources, all experiments initialized both actor and critic models from the widely used Qwen3 series. Evaluation on additional model families (e.g., Llama (Grattafiori et al., 2024)) is left for future work. Following (Liu et al., 2026), we fixed the maximum generation length to 8k tokens, a common academic setting that balances inference coverage while avoiding inference-cost blowups. We plan to assess the algorithm's generalization under ultra-long inference budgets and adopt classical RL practice of using a more diverse set of random seeds to further strengthen the robustness of our conclusions. `AsyPPO` opens new avenues for RL4LLM design and raises several interesting questions. For example, do ensemble critic systems composed of different model families and sizes exhibit performance differences? Do variations in critic hyperparameter settings affect calibration and uncertainty estimates? Promising directions also include confidence-weighted ensemble critics to improve value estimation and analyze the relationship between value uncertainty and entropy.

## ACKNOWLEDGMENTS

The work of Jiashun Liu and Ling Pan is supported by the National Natural Science Foundation of China 62406266. The LLM training is supported by ROLL (Wang et al., 2025c). We would also like to thank the Python community (Van Rossum & Drake Jr, 1995; Oliphant, 2007) for developing tools that enabled this work, including NumPy (Harris et al., 2020), Matplotlib (Hunter, 2007), Jupyter (Kluyver et al., 2016), and Pandas (McKinney, 2013).

ETHICS STATEMENT

This paper presents work whose goal is to advance the field of Machine Learning. There are many potential societal consequences of our work, none which we feel must be specifically highlighted here.

REPRODUCIBILITY STATEMENT

We provide all the details to reproduce our results in the Appendix.

LLM USE

LLMs were used to assist paper editing and to write the code for plotting experiments.

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

# A    DETAILED EXPERIMENTAL SETUP

## A.1    PLOT SETUP

To ensure clarity and intuitiveness in the qualitative analysis, all curves are consistently smoothed using identical parameters. Specifically, the mean values are computed using an 11-step moving window with an exponential smoothing factor of $0.6$. The smooth window set as $4$ and $2$.

## A.2    PROMPT

In this work, we incorporate the following instruction into the system prompt to encourage the model to better demonstrate its reasoning process: **"Please reason step by step, and put your final answer within \boxed{}."** This setting is designed to guide the model to perform step-by-step reasoning and explicitly present the final answer in the form of \boxed{}, thereby enhancing the clarity and readability of the output.

## A.3    HYPERPARAMETERS

We employ ROLL, a user-friendly and efficient open-source reinforcement learning framework, to implement our pipeline. Subsequently, the key parameters observed during the training process are presented as follows. See our code config file for more details on the parameters. For the 14b policy training. We uniformly arrange the actors on (0,16) and the critics on (16,32) GPUs. For other small models, we uniformly place the actor at (0,8) and the critic at (8,16) GPU. Detailed settings can be found in next page.

```
# We use below setup for 4b and 8b policy
seed: 42
max_steps: 500
save_steps: 500
logging_steps: 1
eval_steps: 1
gamma: 1.0  # discount factor
lambd: 1.0  # GAE lambda
rollout_batch_size: 64
prompt_length: 1024
response_length: 8000
value_aggregation_strategy: "mean"
gradient_mask_percentage: 0.2 # mask 20%
entropy_loss_coef: 0.01
entropy_filter_mask_percentage: 0.2 # filter out 20%
ppo_epochs: 1 # 4 is also used in main experiments
adv_estimator: "gae"
init_kl_coef: 0.0
async_generate_level: 1
actor_train:
  training_args:
    learning_rate: 1.0e-6
    weight_decay: 0
    per_device_train_batch_size: 1
    gradient_accumulation_steps: 256
    warmup_steps: 50
    num_train_epochs: 50
```

```
critic_1:
  training_args:
    learning_rate: 1.0e-5
    weight_decay: 1.0e-2
    warmup_steps: 5
    per_device_train_batch_size: 1
    gradient_accumulation_steps: 128
    warmup_steps: 5
    num_train_epochs: 50
critic_2:
  training_args:
    learning_rate: 1.0e-5
    weight_decay: 1.0e-2
    warmup_steps: 5
    per_device_train_batch_size: 1
    gradient_accumulation_steps: 128
    warmup_steps: 5
    num_train_epochs: 50
  ...
actor_infer:
  generating_args:
    max_new_tokens: ${response_length}
    top_p: 0.99
    top_k: 100
    num_beams: 1
    temperature: 0.99
    num_return_sequences: 32
  ...
```

```
# We use below setup for 14b policy
seed: 42
max_steps: 500
save_steps: 500
logging_steps: 1
eval_steps: 1
value_aggregation_strategy: "mean"
gradient_mask_percentage: 0.2 # mask 20%
entropy_loss_coef: 0.01
entropy_filter_mask_percentage: 0.2  # filter out 20% or 0%
rollout_batch_size: 64
prompt_length: 1024
response_length: 8000
infer batch size: 4
ppo_epochs: 4
adv_estimator: "gae"
init_kl_coef: 0.0
async_generate_level: 1
```

```
actor_train:
  training_args:
    learning_rate: 1.0e-6
    weight_decay: 0
    per_device_train_batch_size: 8
    gradient_accumulation_steps: 64
    warmup_steps: 50
    num_train_epochs: 50
critic_1:
  training_args:
    learning_rate: 1.0e-5
    weight_decay: 1.0e-2
    warmup_steps: 5
    per_device_train_batch_size: 2
    gradient_accumulation_steps: 16
    warmup_steps: 5
    infer batch size: 4
    num_train_epochs: 50
critic_2:
  training_args:
    learning_rate: 1.0e-5
    weight_decay: 1.0e-2
    warmup_steps: 5
    per_device_train_batch_size: 2
    gradient_accumulation_steps: 16
    warmup_steps: 5
    infer batch size: 4
    num_train_epochs: 50
  ...
actor_infer:
  generating_args:
    max_new_tokens: ${response_length}
    top_p: 0.99
    top_k: 100
    num_beams: 1
    temperature: 0.99
    num_return_sequences: 32
  ...
```

## B  THE RELATIONSHIP BETWEEN VALUE STD AND STATE INFORMATION QUANTITY

Specifically, for the training scenarios of 8b actors and two 0.6b critics, we use the value-std corresponding to the global state and the median of the gradient magnitude to categorize the states into four types. Namely, large gradient & large value std, large gradient & small value std, small gradient & large value std, small gradient & small value std.

The results in Figure 11 (Left) show that the vast majority of states are classified into the categories of large gradient & large value std and small gradient & small value std, thereby empirically proving the positive relationship between value std and the learning value (information quantity) of the state.

## C  VISUALIZATION OF WORD CLOUDS

We statistically analyzed the word clouds of the tokens with the highest mask frequency in the initial stage of `AsyPPO` training. The results in Figure 11 (Right) show that our mask mechanism tends to mask adjectives, adverbs, and some isolated symbols, with less involvement in logical transitions, except for the slightly prominent progressive word "therefore".

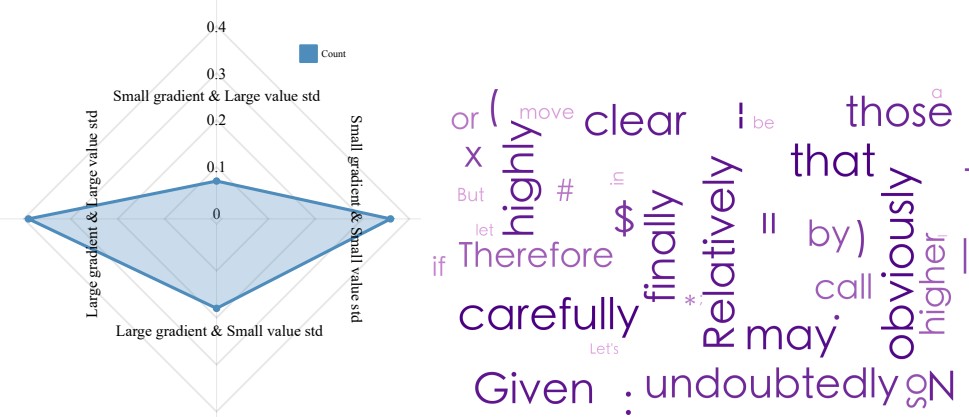

Figure 11: Left: Statistics within a mini-batch in the mid-training stage. Right: The 40 tokens that are masked most frequently in the same mini-batch.

## D  ADDITIONAL RESULTS

**Extensive comparison.** We expanded our comparisons in the revised manuscript and included two strong recent baselines to further understand the effectiveness of our method. Additional baselines: VAPO(Yue et al., 2025): a popular critic-centric RLVR method (evaluated with its default configuration). Q-function Reward Model (Q-RM) + PPO (Chen et al., 2025b): a novel token-level reward method for math reasoning tasks. Experimental setup: To ensure fair comparison, we follow the main experimental protocol using DeepScaleR 40k, with 8k token generation length and 600 training steps, using Qwen3-14B-Base as the policy model. We report the average Avg@4 score across four benchmarks. As shown in Table 1: Compared to critic-centric VAPO, AsyPPO (14B actor, 2×1.7B critic) delivers about 6.5% improvement (while using significantly smaller critic). Compared to the token-level reward model (Q-RM)+PPO, AsyPPO shows about 4% advantage despite Q-RM requiring additional pre-training cost.

| Method | Average Accuracy Across Benchmarks |
|---|---|
| AsyPPO | 61.3 |
| VAPO | 54.8 |
| Q-RM + PPO | 56.2 |

Table 1: Performance comparison.

**Reward sparsity analysis.** We use the process reward model to provide step-wise dense rewards for the policy following, in addition to the extremely sparse reward setting with only the final reward used in the main experiment. The results in Table 2 demonstrate that while both methods benefit from dense rewards, AsyPPO consistently outperforms PPO in both settings.

| Method | AsyPPO | PPO |
|---|---|---|
| Dense Reward | 84.9 | 81.5 |
| Sparse Reward | 83.6 | 78.3 |

Table 2: Performance comparison.

**Domain transfer to code generation.** we added a code-domain evaluation on LiveCodeBench base on DeepSeek-R1-Distill-Qwen-7B, Max Response Len = 8K tokens. As shown in Table 3, AsyPPO consistently outperforms PPO.

| Method | Base Model | After AsyPPO | After PPO |
|---|---|---|---|
| Performance | 45.6 | 55.2 | 49.7 |

Table 3: Performance comparison.

**Validation across multiple model families.**   We have added another series of open-source models that are widely used in academic research. We evaluated Llama-3.1-8B-Base and reported the average score across six benchmarks (Avg@4),i.e., AIME 24, AMC23, GSM8k, MATH-500, Minerva Math, OlypiadBench. As shown in Table 4, AsyPPO achieves an average performance improvement of about 5% over PPO, further demonstrating the effectiveness and generality of our method.

| Model Family | Initial Score | After AsyPPO | After PPO |
|---|---|---|---|
| Qwen3-8B-Base | 41.4 | 59.8 | 57.1 |
| Llama-3.1-8B-Base | 4.6 | 23.8 | 19.2 |

Table 4: Performance comparison.

To assess whether entropy filtering removes rare but valuable states, we analyze the filtered tokens at training steps 100, 200, and 300. Specifically, we compute the proportion of filtered tokens associated with exploration, using the key exploration-related tokens identified in Wang et al. (2025a). As shown in Table 5, the fraction of reasoning-related tokens among the 20% filtered by entropy in the main experiment is very small, indicating that the mechanism does not discard valuable reasoning paths.

| Method | 100 steps | 200 steps | 300 steps |
|---|---|---|---|
| Filter-out ratio (%) | 4.63 | 2.26 | 1.81 |

Table 5: Average results over 16 runs on MATH-500.

**Mitigating rigid advantage calculation.**   AsyPPO employs group-based rollouts, enabling each critic to observe multiple trajectories per prompt. Combined with our non-overlapping data partitioning, critics capture complementary views of the reward distribution, reducing variance in token-level advantage estimates. As shown in Table 6, AsyPPO's advantages are more conservative than GRPO's. This reduces overestimation of frequent non-reasoning phrases in positive samples, e.g., frequently occurring verb phrases, and underestimation of reasoning-critical tokens in negative generations. This prevents collapse into superficial reasoning modes (e.g., repetitive phrases like "So, let's's's...").

| Sample | 100 step | 200 step | 300 step |
|---|---|---|---|
| Positive samples (GRPO: 1) | $0.91 \pm 0.13$ | $0.84 \pm 0.06$ | $0.87 \pm 0.11$ |
| Negative samples (GRPO: 0) | $0.26 \pm 0.35$ | $0.13 \pm 0.18$ | $0.09 \pm 0.16$ |

Table 6: Average token advantage.

