# OpenReview forum: "Asymmetric Proximal Policy Optimization: mini-critics boost LLM reasoning"
_ICLR.cc/2026/Conference — ICLR 2026 Poster_

### Official Review · Reviewer_yexD · 2025-10-31

**Soundness:** 2
**Presentation:** 3
**Contribution:** 2
**Rating:** 4
**Confidence:** 2

**Summary:**

This paper introduces Asymmetric Proximal Policy Optimization (AsyPPO), a lightweight actor–critic method for reinforcement learning fine-tuning of large language models. It employs several small “mini-critics” trained on non-overlapping data to provide ensemble value estimates. The agreement and disagreement between critics are used to stabilize policy updates through masking and filtering. Experiments on multiple reasoning benchmarks show 3–6% accuracy gains over PPO and GRPO with lower memory usage and faster training.

**Strengths:**

1.This paper is well motivated.
The paper revisits a central assumption of modern RL4LLM, namely the removal of explicit critics (as in GRPO), and proposes an elegant asymmetric redesign that restores the critic’s role while remaining scalable.

2.This paper is easy to understand.
The proposed mini-critic ensemble, prompt-level non-overlapping partition, and uncertainty-based masking/filtering are clearly described and intuitively justified.

3. Strong exps results.
The method consistently improves performance across several benchmarks, while reducing memory footprint by ~20% and maintaining comparable speed to GRPO.

**Weaknesses:**

1. Limited interpretability of critic signals
The mini-critics are trained without access to final-answer correctness; hence their value estimates do not guarantee semantic or symbolic validity, especially on complex mathematical reasoning tasks. The reported gains may stem from smoother optimization rather than genuinely better reasoning reward estimation.

2. Lack of compatibility study with GRPO framework
Following point 1, as GRPO could provide "correct" outcome reward, it is essential to verify whether AsyPPO can be plugged into GRPO’s group-based rollout and reward pipeline as a complementary.

3. Evaluation scope
All experiments are restricted to the Qwen3 model family and math-style datasets. It remains uncertain whether the observed benefits generalize to more diverse reasoning or open-ended tasks and models.

**Questions:**

Please response to the points listed in the weaknesses

---

> ### Author Response · Authors · 2025-11-21
> **Rebuttal (Part 1)**
>
> Thanks for your thorough, meticulous, and impartial feedback. It substantially helps strengthen our work. We hope the following responses address your concerns.
>
> ### Weaknesses
>
> > W1: Limited interpretability of critic signals. The mini-critics are trained without access to final-answer correctness; hence their value estimates do not guarantee semantic or symbolic validity, especially on complex mathematical reasoning tasks. The reported gains may stem from smoother optimization rather than genuinely better reasoning reward estimation.
> >
>
> We apologize for the confusion caused by the ambiguous expression. In the new version, we have conducted additional experiments to analyze the mechanism of mini-critics from multiple perspectives.
>
>  **Access to final-answer correctness:** In fact, our critics are explicitly trained to estimate the expected discounted return from each state $s$ by taking action $a$ and following the policy $\pi$ afterwards (according to the definition of the critic/value-function). This is shown in the loss function below, where $R_t$ is directly derived from final answer correctness. Each trajectory reaches a terminal state with a definitive reward signal, providing critics with ground-truth supervision directly linked to solution correctness. We also empirically validated this point by comparing critic value estimates (using the checkpoint from step 150 in the main experiment of AsyPPO (14b actor and 1.7b critic) with actual rewards on MATH -500. The minimal difference in Table 1 confirms that critics successfully learn to estimate solution correctness.
>
> $$\mathcal{L}{\text{critic}}(\boldsymbol{\phi}) =
> \sum^M_{m=1} \mathcal{L}{\text{critic}}^{(m)}(\phi_m) =
> \sum^M_{m=1} \mathbb{E}_{(s_t, R_t) \sim \mathcal{D}_m} \left[
> \left( V(s_t; \phi_m) - R_t \right)^2
> \right] $$
>
> Table 1. Critic estimates accuracy
>
> | Method | The difference from the final reward |
> | --- | --- |
> | AsyPPO | $0.08\pm0.13$ |
>
> **Mitigating rigid advantage calculation:** AsyPPO employs *group-based rollouts*, enabling each critic to observe multiple trajectories per prompt. Combined with our non-overlapping data partitioning, critics capture complementary views of the reward distribution, reducing variance in token-level advantage estimates. As shown in Table 2, AsyPPO’s advantages are more conservative than GRPO’s. This reduces overestimation of frequent non-reasoning phrases in positive samples, e.g., frequently occurring verb phrases, and underestimation of reasoning-critical tokens in negative generations. This prevents collapse into superficial reasoning modes (e.g., repetitive phrases like *“So, let's’s’s…”*).
>
> Table 2. Average token advantage
> | Sample | 100 step | 200 step | 300 step |
> | --- | --- | --- | --- |
> | Positive samples (GRPO: 1) | $0.91\pm0.13$ | $0.84\pm0.06$ | $0.87\pm0.11$ |
> | Negative samples  (GRPO: 0) | $0.26\pm0.35$ | $0.13\pm0.18$ | $0.09\pm0.16$ |
>
> **Smooth Policy updating:** Table 3 presents the policy gradient magnitude analysis for GRPO and AsyPPO (14b actor, 2*1.7b critic). The results indicate that the token-level value calculations based on the critic produce smoother gradient magnitude for the policy. This smoothness may help maintain the stability of policy optimization.
>
> Table 3. Average gradient magnitude during the training phase
>
> | Method | Gradiant magitude |
> | --- | --- |
> | AsyPPO | $0.27\pm0.06$ |
> | PPO | $0.79\pm 1.47$ |

---

> ### Author Response · Authors · 2025-11-21
> **Rebuttal (Part 2)**
>
> > W2: Lack of compatibility study with GRPO framework. Following point 1, as GRPO could provide "correct" outcome reward, it is essential to verify whether AsyPPO can be plugged into GRPO’s group-based rollout and reward pipeline as a complement.
> >
>
> Thank you for this suggestion! Our method design is general and can be built upon GRPO. In the new version, we systematically analyze how AsyPPO's techniques can complement GRPO.
>
> **GRPO with our mini-critics' advantage calculation:**  We integrate two mini-critics with Qwen3-1.7b-Base into the GRPO framework. These mini-critics help estimate the token-level advantage of the GRPO rollout in an AsyPPO style, which replaces GRPO's traditional advantage calculation. As shown in the table below, this integration leads to more stable incremental gains.
>
> Table 5. Average @ 16 on MATH-500
>
> | Method | Peak score |
> | --- | --- |
> | Vanilla GRPO | 80.1 |
> | GRPO w/ mini-critics' advantage calculation | 83.9 |
>
> **GRPO with our advantage masking based on agreement:** We utilize two mini-critics for GRPO (Qwen3-1.7b-Base) and mask tokens with low learning benefits based on the standard deviation between critics during the training phase. Such a combination effectively avoids the token-level gradient noise caused by the advantage calculation at the group level of GRPO. This integration shows improved performance, as in Table 6.
>
> Table 6. Average @ 16 on MATH-500
>
> | Method | Peak score |
> | --- | --- |
> | Vanilla GRPO | 80.1 |
> | GRPO w/ advantage masking based on agreement | 85.2 |
>
> **GRPO with our entropy filtering based on divergence:** We incorporate entropy regularization into the GRPO loss function and employ two mini-critics, specifically Qwen3-1.7b-Base. During entropy calculation, tokens with significant value differences across the critics are filtered out. The results in Table 7 demonstrate that this combination facilitates robust exploration of the policy and achieves better performance.
>
> Table 7. Average @ 16 on MATH-500
>
> | Method | Peak score |
> | --- | --- |
> | Vanilla GRPO | 80.1 |
> | GRPO w/ entropy filtering based on divergence | 84.7 |
>
> > W3: Evaluation scope All experiments are restricted to the Qwen3 model family and math-style datasets. It remains uncertain whether the observed benefits generalize to more diverse reasoning or open-ended tasks and models.
> >
>
> Thanks for the suggestion. We conducted extensive additional experiments demonstrating AsyPPO's effectiveness across different model architectures and tasks to address the concern.
>
> **Generalization across the model families:** We test the cross-model generalization ability of AsyPPO based on Llama-3.1-8b-Base, following the training setup in [1]. The results in  Table 8 indicate that AsyPPO's advantages translate robustly to the Llama architecture, where we observe that AsyPPO's average performance on six math benchmarks is higher than that of  PPO, further validating the effectiveness of AsyPPO.
>
> Table 8. Record the peak score after convergence
>
> | Method | Base model | + AsyPPO | + PPO |
> | --- | --- | --- | --- |
> | Preformance | 4.6 | 23.8 | 19.2 |
>
> **Generalization across tasks:** Following [2], we added a code-domain evaluation on LiveCodeBench based on DeepSeek-R1-Distill-Qwen-7B, with max response length = 8K tokens. As shown in Table 9, AsyPPO consistently outperforms PPO.
>
> Table 9. Record the Pass@8 score
>
> | Method | Base model | + AsyPPO | + PPO |
> | --- | --- | --- | --- |
> | Preformance | 45.6 | 55.2 | 49.7 |
>
> Our additional experiments conclusively demonstrate that AsyPPO's benefits generalize robustly across both model families and task domains.
>
> [1]: SimpleRL-Zoo: Investigating and Taming Zero Reinforcement Learning for Open Base Models in the Wild COLM (2025)
>
> [2]: Zhu, Xuekai et al. “FlowRL: Matching Reward Distributions for LLM Reasoning.” ArXiv abs/2509.15207 (2025)
>
> ---
>
> Thanks for reviewing our work! We hope our responses address the concerns and would appreciate your consideration in the evaluation. We are happy to provide further clarification if needed.

---

> > ### Author Response · Authors · 2025-11-27
> >
> > Dear Reviewer yexD,
> >
> > We hope this message finds you well. As the author-reviewer discussion period concludes in less than a week, we wanted to kindly follow up on our response to your valuable comments and questions.
> >
> > In our rebuttal, we addressed your concerns by theoretically and empirically demonstrating that our critic can assess the correctness of final answers. We showed that our mini-critics can mitigate rigid advantage calculations and provide a smoother policy update signal. Additionally, we conducted further experiments to illustrate that the techniques in AsyPPO also enhance the performance of GRPO. Finally, we evaluated our method's generalization ability from both model and domain perspectives.
> >
> > We believe we have adequately clarified the concerns you raised. If you have any further thoughts or questions regarding our responses, we would be more than happy to address them.
> >
> > Thank you once again for your insightful review! We look forward to your feedback.
> >
> > Best regards,
> >
> > The authors

---

### Official Review · Reviewer_FX1U · 2025-10-31

**Soundness:** 2
**Presentation:** 3
**Contribution:** 2
**Rating:** 4
**Confidence:** 3

**Summary:**

The paper proposes AsyPPO as a lightweight method for incorporating more than one mini-critics in RL. Instead of a full-size critic, AsyPPO uses two lightweight "mini-critics" trained on disjoint, prompt-level shards to encourage diversity while staying calibrated. Their predictions are averaged for value estimation, and the standard deviation across critics is treated as an uncertainty signal: the method masks advantages for states where critics strongly agree (to avoid low-information updates) and excludes high-disagreement states from the entropy term (to avoid spurious exploration). On math-reasoning benchmarks, AsyPPO reports notable average gains over GRPO, classic PPO, and lower peak memory / faster steps than symmetric PPO, while remaining competitive and more lightweight.

**Strengths:**

1. Re-introducing critics via small, prompt-sharded ensembles seems (to me) a clear, practical twist that fits LLM training constraints and mitigates over-parameterized critics. The paper formalizes the training and aggregation cleanly.
2. The two signals, agreement-based advantage masking and divergence-based entropy filtering, are well-motivated and implemented directly inside PPO, with ablations supporting the choices.
3. The paper shows lower peak memory and faster training steps vs. symmetric PPO, and consistent accuracy gains vs. GRPO/symmetric PPO under off-policy reuse. The "two critics is the sweet spot" result is helpful guidance.

**Weaknesses:**

1. Strong critic-free baselines like GRPO are covered, but comparisons to recent critic-centric or token-level reward methods are referenced more than reproduced, making it hard to isolate where wins come from in identical settings.
2. Claims of stability under off-policy (UTD>1) are promising, but it's unclear how performance changes with different UTD, lambda, or reward sparsity beyond the selected setup.
3. Experiment-wise, most results are math-reasoning; model family is Qwen3-only. This raises external validity concerns (other domains, model families like Llama, non-verifiable reward, etc.). I strongly suggest validating the claims on at least some coding benchmarks.
4. The method introduce addition hyperparameters, including the choice and number of critics, value aggregation, advantage masking fraction, and entropy filtering fraction. Without cross-domain experiments, it's challenging to assess how sensitive these hyperparameters are to different domains, tasks, or model families in practice (and if they are easy to tune).

**Questions:**

1. Are masking and entropy filtering applied strictly token-wise or differently?
2. The paper shows gains with bigger critics. Where's the diminishing return point relative to actor size, and how does that trade off with ensemble count? And any insights on why more critics does not yield meaningful return?
3. Can entropy filtering over-prune rare but valuable states (e.g., creative solution paths), reducing long-tail discovery?

---

> ### Author Response · Authors · 2025-11-21
> **Rebuttal (Part 1)**
>
> We sincerely thank you for the detailed feedback and the thoughtful suggestions. We hope the following responses address your concerns.
>
> ### Weakness
>
> > W1: Strong critic-free baselines like GRPO are covered, but comparisons to recent critic-centric or token-level reward methods are referenced more than reproduced, making it hard to isolate where wins come from in identical settings.
> >
>
> Thanks for the valuable suggestion! To address the concern, we  expanded our comparisons in the revised manuscript and included two strong recent baselines  to further understand the effectiveness of our method:
>
> *Additional baselines*: VAPO[1]: a popular critic-centric RLVR method (evaluated with its default configuration). Q-function Reward Model (Q-RM) + PPO [2]: a novel token-level reward method for math reasoning tasks.
>
> Experimental setup: To ensure fair comparison, we follow the main experimental protocol using DeepScaleR 40k, with 8k token generation length and 600 training steps, using Qwen3-14B-Base as the policy model. We report the average Avg@4 score across all six benchmarks.
>
> Results: As shown in Table 1:
>
> - Compared to critic-centric VAPO, AsyPPO (14B actor, 2×1.7B critic) delivers about **6.5%** improvement (while using significantly smaller critic).
>
> - Compared to the token-level reward model (Q-RM)+PPO,  AsyPPO shows  **> 4%** advantage despite Q-RM requiring additional pre-training cost.
>
> These additional comparisons show that our approach consistently outperforms both critic-centric and token-level reward methods, confirming that the architectural innovation and the masking mechanism design are the primary source of improvement (further supported by ablations in Section 3.2).
>
> Table 1. Performance comparison
>
> | Method | Average accuracy across six benchmarks |
> | --- | --- |
> | **AsyPPO** | 61.3 |
> | VAPO | 54.8 |
> | Q-RM + PPO | 56.2 |
>
> [1]: Yue, Yu, et al. "Vapo: Efficient and reliable reinforcement learning for advanced reasoning tasks." arXiv:2504.05118 (2025).
>
> [2]: Chen, Hongzhan, et al. "Discriminative Policy Optimization for Token-Level Reward Models." ICML (2025).

---

> ### Author Response · Authors · 2025-11-21
> **Rebuttal (Part 2)**
>
> > W2: Claims of stability under off-policy (UTD>1) are promising, but it's unclear how performance changes with different UTD, lambda, or reward sparsity beyond the selected setup.
>
> In the revised version, we add extensive additional experiments investigating AsyPPO (Qwen3-8B-Base actor, 2xQwen3-1.7B-Base critic)'s stability under varying conditions. We conduct fair evaluations on the widely used MATH-500 benchmark, report  Avg@16 metric (to mitigate contingency), and train for 800 steps.
>
> **Stability across UTD values:** Table 2 evaluates AsyPPO with various UTD ratios (ranging from 1 to 6). The results demonstrate that AsyPPO remains effective across UTD values, with performance improving modestly as UTD increases up to 4 (with higher data reuse). At UTD=6, performance slightly declines, likely due to instability from off-policy drift (the growing mismatch between the training policy and the sample policy) which aligns with theoretical expectations for off-policy learning [1].
>
> Overall, these findings confirm that AsyPPO is stable in the widely adopted UTD range, and further reducing instability from distribution mismatch is an interesting direction for future work.
>
> Table 2. Performance with different UTD values
>
> | Method | AsyPPO |
> | --- | --- |
> | UTD=1 | 81.3 |
> | UTD=2 | 81.6 |
> | **UTD=4** | 83.1 |
> | UTD=6 | 79.8 |
>
> **GAE-$\lambda$ sensitivity analysis:** Table 3 reports a sweep of the GAE-$\lambda$ parameter over {0.97, 0.98, 0.99, 1.0} under a unified setup. In RLVR-style settings, popular methods like  PPO typically use Monte Carlo returns ($\lambda = 1$), which can become unstable for smaller $\lambda$ values. Notably, we find that AsyPPO demonstrates robust performance for $\lambda$ values in the range 0.98-1.0, with only a minor decline. We only observe a modest drop at $\lambda=0.97$, indicating the boundary of stability for this parameter.
>
> Table 3. Performance with Different $\lambda$ values
>
> | Method | AsyPPO |
> | --- | --- |
> | $\lambda$=1 | 83.1 |
> | $\lambda$=0.99 | 82.7 |
> | $\lambda$=0.98 | 82.1 |
> | $\lambda$=0.97 | 76.2 |
>
> **Reward sparsity analysis:**  We use the process reward model to provide step-wise dense rewards for the policy following [2], in addition to the extremely sparse reward setting with only the final reward used in the main experiment.  The results in Table 4 demonstrate that while both methods benefit from dense rewards, AsyPPO consistently outperforms PPO in both settings.
>
> Table 4. Performance with dense vs. sparse rewards
>
> | Method | AsyPPO | PPO |
> | --- | --- | --- |
> | Dense reward | 84.9 | 81.5 |
> | Sparse reward | 83.6 | 78.3 |
>
> Furthermore, we consider training data with different difficulty levels and sizes to investigate the sparse level in terms of the reward landscape.  Specifically, we evaluate the performance of PPO and AsyPPO using Lite PPO-Hard[3], a smaller-scale (5k) and easier dataset, and DeepscaleR[4], a larger-scale (40k) and more difficult one, respectively. The results in Table 5 show that AsyPPO outperforms PPO on datasets with different difficulty levels (corresponding to different sparsity levels in the reward landscape).
>
> Table 5. Record the peak score after convergence
>
> | Method | AsyPPO | PPO |
> | --- | --- | --- |
> | DeepScaleR | 85.1 | 82.2 |
> | LitePPO-hard | 83.6 | 78.3 |
>
> [1]: Meng, Wenjia et al. “Off-Policy Proximal Policy Optimization.” *AAAI Conference on Artificial Intelligence* (2023).
>
> [2]: Cui, Ganqu et al. “Process Reinforcement through Implicit Rewards.” *ArXiv* abs/2502.01456 (2025): n. pag.
>
> [3]: Liu, Zihe et al. “Part I: Tricks or Traps? A Deep Dive into RL for LLM Reasoning.” *ArXiv* abs/2508.08221 (2025)
>
> [4]: huggingface.co/datasets/agentica-org/DeepScaleR-Preview-Dataset

---

> > ### Author Response · Authors · 2025-11-21
> > **Rebuttal ( Part 3)**
> >
> > > W3: Experiment-wise, most results are math-reasoning; model family is Qwen3-only. This raises external validity concerns (other domains, model families like Llama, non-verifiable reward, etc.). I strongly suggest validating the claims on at least some coding benchmarks.
> > >
> >
> > In the new version, we have expanded our evaluation to address the concern.
> >
> > **(Validation across multiple model families)**
> >
> > We have added another series of open-source models that are widely used in academic research.Following the training setup in [1], we evaluated Llama-3.1-8B-Base and reported the average score across six benchmarks (Avg@4),i.e., AIME 24, AMC23, GSM8k, MATH-500, Minerva Math, OlypiadBench.
> >
> > As shown in Table 5, AsyPPO achieves an average performance improvement of about **5%** over PPO, further demonstrating the effectiveness and generality of our method.
> >
> > Table 5. Cross-Model Family performance
> >
> > | Model Family | Initial score | After AsyPPO | After PPO |
> > | --- | --- | --- | --- |
> > | Qwen3-8b-Base | 41.4 | 59.8 | 57.3 |
> > | Llama-3.1-8b-Base | 4.6 | 23.8 | 19.2 |
> >
> > **(Domain transfer to code generation)**
> >
> > Following [2], we added a code-domain evaluation on LiveCodeBench base on DeepSeek-R1-Distill-Qwen-7B, Max Response Len = 8K tokens. As shown in Table 6, AsyPPO consistently outperforms PPO.
> >
> > Table 6.  Code generation performance (pass@8 on LiveCodeBench)
> >
> > | Method | Base model | After AsyPPO | After PPO |
> > | --- | --- | --- | --- |
> > | Preformance | 45.6 | 55.2 | 49.7 |
> >
> > [1]: SimpleRL-Zoo: Investigating and Taming Zero Reinforcement Learning for Open Base Models in the Wild COLM (2025)
> >
> > [2]: Zhu, Xuekai et al. “FlowRL: Matching Reward Distributions for LLM Reasoning.” ArXiv abs/2509.15207 (2025)
> >
> > > W4: The method introduces additional hyperparameters, including the choice and number of critics, value aggregation, advantage masking fraction, and entropy filtering fraction. Without cross-domain experiments, it's challenging to assess how sensitive these hyperparameters are to different domains, tasks, or model families in practice (and if they are easy to tune).
> > >
> >
> > We analyzed the robustness of AsyPPO’s parameter configurations by evaluating the trained policy (checkpoint of AsyPPO-8b actor-2x1.7b critic and GRPO-8b) on a widely used out-of-domain task, i.e., GPQA. GPQA [1] is a scientific question-and-answer dataset containing 448 multiple-choice questionscovering subfields of biology, physics and chemistry. As shown in  Table 7, AsyPPO with these default settings outperforms GRPO, indicating reasonable cross-domain robustness.
> >
> > Table 7. Cross-Domain performance on GPQA (avg@5)
> >
> > | Method | Base model | AsyPPO | GRPO |
> > | --- | --- | --- | --- |
> > | Preformance | 9.4 | 39.6 | 36.1 |
> >
> > Due to limited GPUs, we will continue to evaluate AsyPPO's OOD performance on various tasks , e.g., open-ended question answering, puzzle tasks.
> >
> > [1]: Rein, David, et al. "Gpqa: A graduate-level google-proof q&a benchmark." COLM. 2024
> >
> > ### Questions
> >
> > > **Q1:** Are masking and entropy filtering applied strictly token-wise or differently?
> >
> > In the new version, we included clarifications that both masking and entropy filtering are applied strictly token-wise.

---

> ### Author Response · Authors · 2025-11-21
> **Rebuttal (Part 4)**
>
> > Q2: The paper shows gains with bigger critics. Where's the diminishing return point relative to actor size, and how does that trade off with ensemble count? And any insights on why more critics does not yield meaningful returns?
> >
>
> **Analysis of critic size scaling:** Following the main experimental setup, we further evaluate the impact of different critic sizes on performance with various sizes of actors (Qwen3-8b-base, Qwen3-14b-base).
>
> The results in Table 8 show that, for both actor sizes, performance continues to improve as critic size increases from 1.7B to 8B. However, the performance of 8B and 14B critics is nearly identical, **suggesting that an 8B critic offers an effective trade-off between computational cost and performance** for both the 8B actor used in the original setup and the 14B actor examined in the extended experiments.
>
> Table 8. Impact of critic size on performance (MATH-500, avg@16)
>
> | Critic size | 8b actor | 14b actor |
> | --- | --- | --- |
> | 1.7b critic | 71.5 | 81.2 |
> | 4b critic | 74.1 | 82.7 |
> | 8b critic | 74.7 | 83.4 |
> | 14b critic | 74.4 | 83.6 |
>
> **Analysis of the number of critics:** We use the 8B critic recommended in the previous evaluation to examine how the number of critics affects performance across two types of actors. The results presented in Table 9 indicate that two critics achieve optimal performance with actors of varying sizes. This aligns with our conclusion: **for policy models of different sizes, two critics demonstrate good performance.**
>
> Table 9. Record the average @ 16 on MATH-500
>
> | Method | 8b actor | 14b actor |
> | --- | --- | --- |
> | n=1 | 67.6 | 77.2 |
> | n=2 | 71.3 | 83.4 |
> | n=4 | 71.1 | 83.1 |
>
> Analysis of why increasing the number of critics has not provided meaningful returns:
>
> - The generations under a single prompt are not extremely diverse. Therefore, two critics are sufficient for effective semantic representation. As task complexity increases, more critics may demonstrate better performance.
> - The differences among critics inspired by our group-level non-overlap data division gradually decrease during the later stages of training, leading to the inability of critics to gain further benefits.
>
> Research insights: A promising future direction is to *explicitly* encourage critic differentiation. This could be achieved by designing distinct optimization objectives for each critic, or by assigning **different λ values** to integrate value estimates across different advantage-calculation horizons (e.g., λ=1.0 for Critic A and λ=0.95 for Critic B). Such heterogeneous critics may improve robustness and better leverage ensemble effects.
>
> > Q3: Can entropy filtering over-prune rare but valuable states (e.g., creative solution paths), reducing long-tail discovery?
> >
>
> To assess whether entropy filtering removes rare but valuable states, we analyze the filtered tokens at training steps 100, 200, and 300. Specifically, we compute the proportion of filtered tokens associated with exploration, using the key exploration-related tokens identified in [1]. As shown in Table 10, the fraction of reasoning-related tokens among the 20% filtered by entropy in the main experiment is **very small**, indicating that the mechanism does not discard valuable reasoning paths.
>
> Table 10. Record the average @ 16 on MATH-500
>
> | Method | 100 step | 200 step | 300 step |
> | --- | --- | --- | --- |
> | filter out ratio （%） | 4.63 | 2.26 | 1.81 |
>
> In addition,  Figure 10 in the manuscript also shows that the filtered high-frequency tokens are mainly modal words or adverbs unrelated to inference, and rarely involve fork tokens.
>
> [1]:  Wang, Shenzhi, et al. "Beyond the 80/20 rule: High-entropy minority tokens drive effective reinforcement learning for llm reasoning." NeuIPS (2025).
>
> [2]: He, Haoran, et al. "Random Policy Valuation is Enough for LLM Reasoning with Verifiable Rewards." *arXiv:2509.24981* (2025).
>
> ---
>
> Thanks for reviewing our work! We hope our responses address the concerns, and we appreciate your consideration in the evaluation. We are happy to provide further clarification if needed.

---

> > ### Comment · Reviewer_FX1U · 2025-11-25
> >
> > I appreciate the authors' efforts during the rebuttal and the additional experiments that addressed my concerns. I have carefully reviewed the results above and gone through the revised pdf. I am generally positive about the paper and have changed my score from 4 to 6.

---

> > ### Public Comment · ~Merak_P1 · 2026-04-30
> > **Inconsistency between Table 8 and Table 9 in the rebuttal**
> >
> > Both tables seem to report results on the same configuration of 8B actor + 8B critic:
> >
> > Table 8 (critic size scaling, presumably with n=2 critics following the main setup): 8B actor + 8B critic = 74.7
> > Table 9 (number of critics, using "the 8B critic recommended in the previous evaluation"): 8B actor + n=2 (i.e., 2×8B critic) = 71.3
> > The 14B actor row matches exactly across both tables (83.4), suggesting these are intended to be the same experimental setup. Yet the 8B actor row differs by 3.4 points (74.7 vs 71.3).
> >
> > Could you clarify:
> >
> > Are the two configurations actually identical? If Table 8 uses a different n (e.g., n=1), please note that this would conflict with Table 9's n=1 result of 67.6 — which would in turn raise concerns about Table 9's monotonicity.
> > If they are the same configuration, what causes the 3.4-point gap? Is it run-to-run variance from different seeds, different training checkpoints, or different evaluation snapshots?
> > If the gap is attributable to seed variance, this would be concerning because it is comparable in magnitude to the ~3-point improvement AsyPPO claims over GRPO on 8B/14B actors. Could you provide error bars or multi-seed results to demonstrate that the reported gains are statistically meaningful rather than within the noise floor of single-seed runs?
> > This clarification would significantly strengthen the empirical claims of the paper.

---

> ### Author Response · Authors · 2025-11-25
> **Thanks for your recognition of our contributions and for supporting the acceptance of our submission!**
>
> We are glad we have addressed your concerns, and thank the reviewer for revising their score!
>
> If there is anything else we can address that would make the reviewer more strongly supportive of our work, please let us know.

---

### Official Review · Reviewer_AgbJ · 2025-11-01

**Soundness:** 3
**Presentation:** 4
**Contribution:** 3
**Rating:** 6
**Confidence:** 3

**Summary:**

This paper introduces Asymmetric Proximal Policy Optimization (AsyPPO), a framework designed to address the high computational cost and instability of training large critic models in Reinforcement Learning for LLMs (RL4LLM). Instead of the conventional symmetric design where the critic is as large as the actor, AsyPPO utilizes an ensemble of lightweight "mini-critics". To ensure these small critics provide robust and diverse guidance, they are trained on disjoint, non-overlapping data partitions. The framework further leverages the uncertainty between these critics to refine the policy objective: it masks advantages in states where critics agree (low uncertainty) to prevent overfitting to low-information samples, and it filters states where critics diverge (high uncertainty) from the entropy regularization term to promote safer exploration. The authors demonstrate through experiments that AsyPPO improves reasoning performance over baselines like GRPO and achieves significant reductions in computational overhead (memory and time) compared to classic symmetric PPO.

**Strengths:**

- The paper addresses the significant and timely problem of the computational bottleneck in critic-based RL for post-training LLMs
- The proposed method is intuitive and well-motivated.
- The paper is well-written, easy to follow, and provides a thorough analysis of each of its components, supported by targeted ablation studies (e.g., ensemble strategy, advantage masking, and entropy filtering).
- The method is validated across a variety of LLM reasoning benchmarks.
- The literature review is comprehensive and effectively situates the work within the current RL4LLM landscape.
- The authors include a clear discussion of the work's limitations and potential avenues for future research.

**Weaknesses:**

- The empirical evaluations could be strengthened. Several learning curves in the figures do not appear to have fully stabilized or converged, showing volatility late in training (e.g., in Figure 4(c), Figure 6(b), and parts of Figure 7). Extending the training steps could provide a clearer validation of the method's stability and final performance.

- The performance gains, while consistent, are not always substantial. For instance, the main results in Figure 7 show an average improvement of "about 3 points" over the GRPO baseline, which may be modest.

- The study's baselines could be more comprehensive. While GRPO is a strong baseline, the paper would benefit from comparison against other advanced RL4LLM algorithms mentioned in the related work, such as DAPO or REINFORCE++, to better contextualize its contributions.

**Questions:**

- The paper frequently uses the term "UTD" (update-to-data ratio), setting it to 4 in key experiments. The appendix also lists `ppo_epochs` as a hyperparameter, setting it to 1 or 4. Could the authors explicitly clarify the definition of UTD and explain the precise difference between it and the `ppo_epochs` hyperparameter in their training setup?

---

> ### Author Response · Authors · 2025-11-21
> **Rebuttal (Part 1)**
>
> We are immensely grateful to you for recognizing the importance and value of our research. Your suggestions inspire us to improve our work further.
>
> ### Weakness
>
> > W1: The empirical evaluations could be strengthened. Several learning curves in the figures do not appear to have fully stabilized or converged,..Extending the training steps could provide a clearer validation of the method's stability and final performance.
>
> To address this concern (which was due to limited resources), we have further extended our experiments from 400 to 600 training steps (demonstrating clearer stability and convergence patterns) with the original experimental setup (considering the key baselines for ablations in Figs. 4(c) and 6(b), and performance comparison in Fig. 7). The following tables summarize key outcomes from the extended runs, which strengthen our claims. We will update all figures in our final submission.
>
> **Advantage masking based on value agreement (Fig. 4(c))**
>
> The results in Table 1 confirm that our masking mechanism based on value agreement yields stable performance improvements with a significant gap of approximately 8% after convergence. This indicates that our advantage masking scheme effectively mitigates overfitting during continual training.
>
> Table 1. Final performance
>
> | Method | Average accuracy across six benchmarks |
> | --- | --- |
> | **AsyPPO w/ advantage masking** | 54.3 |
> | AsyPPO w/o advantage masking | 46.4 |
>
> **Entropy filtering based on value divergence (Fig. 6(b))**
>
> The extended results in Table 2 show that our entropy-filtering method delays premature convergence, yielding a stable improvement of > 5%.
>
> Table 2. Final performance
>
> | Method | Average accuracy across six benchmarks |
> | --- | --- |
> | **AsyPPO w/ entropy filtering** | 49.1 |
> | AsyPPO w/o entropy filtering | 43.6 |
>
> **Comparison with GRPO (Fig. 7)**
>
> Extended training (with 14B actor model) further reinforces our findings:
>
> - AsyPPO (2×4B critic) consistently outperforms GRPO by > 6% across benchmarks, where the performance advantage remains stable after convergence.
> - Our AsyPPO (2×4B critic) outperforms our AsyPPO (2×1.7B critic) by about 2%, supporting our finding that using the largest critic model that fits in GPU memory maximizes AsyPPO’s optimization capacity.
>
> Table 3. Final converged performance
>
> | Method | Average accuracy across six benchmarks |
> | --- | --- |
> | **AsyPPO (2×4B critic)** | 64.6 |
> | AsyPPO (2×1.7B critic) | 62.3 |
> | GRPO | 58.1 |
>
> These extended results strengthen the stability and final-performance claims without altering our original conclusions.
>
> > W2: The performance gains, while consistent, are not always substantial. For instance, the main results in Figure 7 show an average improvement of "about 3 points" over the GRPO baseline, which may be modest.
>
> ### Additional experiments show more substantial gains
>
> We  address this concern through additional experimental evidence that better evaluates AsyPPO on the 14B policy model with two additional validations:
>
> *Extended training* (same setup as in Fig. 7): As detailed in our response to **W1 (Table 3)**, extending training from 400 to 800 steps reveals a larger and more stable performance gap. AsyPPO (2×4B) outperforms GRPO by 6.5% on the main evaluation metric across benchmarks, attributable to our design for more effective exploration and filtering of low-quality gradients, demonstrating that the full advantages of our method emerge with sufficient training time.
>
> *Scaling the training dataset:* We additionally scale the training set from 5k to the DeepScaleR 40k dataset [1] while keeping all other settings unchanged (600 training steps). The resulting 9.6% improvement shown in Table 4 demonstrates that AsyPPO leverages larger, more diverse datasets considerably more effectively than baseline approaches. This is particularly relevant as training dataset sizes continue to grow in practical applications.
>
> Table 4. Results on the DeepScaleR dataset
>
> | Method | Average accuracy across six benchmarks |
> | --- | --- |
> | **AsyPPO w/ AsyPPO (2×4B critic)** | 65.7 |
> | GRPO | 56.1 |
>
> ### Beyond performance: key contributions of AsyPPO
>
> Although performance improvements are important, and improve substantially under extended training and larger datasets,  AsyPPO’s key contributions go beyond benchmark numbers:
>
> **Resource Efficiency**: AsyPPO ~30% lighter than symmetric PPO while achieving superior results, enabling efficient RLVR in resource-constrained settings.
>
> - **Architectural Innovation & Insight**: Our approach challenges the traditional assumption in classic RL that actor and critic should have symmetric capacity, opening new design possibilities for RL with pre-trained foundation models. We provide new insights into when critics benefit RL with pretrained models, offering design guidance for future work.
> - **Practical Applicability**:  AsyPPO enables effective LLM alignment at scales that would be prohibitively expensive to train under symmetric PPO.

---

> > ### Author Response · Authors · 2025-11-21
> > **Rebuttal (Part 2)**
> >
> > > W3: The study's baselines could be more comprehensive. While GRPO is a strong baseline, the paper would benefit from comparison against other advanced RL4LLM algorithms mentioned in the related work, such as DAPO or REINFORCE++, to better contextualize its contributions.
> > >
> >
> > Thanks for this suggestion. In the new version, we implement and evaluate **REINFORCE++** as an additional baseline, using its default settings. AsyPPO outperforms REINFORCE++ by **~5%** under the main experimental setup (Table 5), further validating the effectiveness of our approach.
> >
> > Due to resource constraints, we could not implement all possible baselines. However, the masking mechanisms introduced in AsyPPO can be integrated into DAPO or other related frameworks, and we plan to explore this in follow-up work.
> >
> > Table 5. Performance comparison with REINFORCE++
> >
> > | Method | Average accuracy across six benchmarks |
> > | --- | --- |
> > | **AsyPPO w/ AsyPPO (2×4B critic)** | $66.3$ |
> > | REINFORCE++ | $61.4$ |
> >
> > ### Question
> >
> > > Q1: The paper frequently uses the term "UTD" (update-to-data ratio), setting it to 4 in key experiments. The appendix also lists ppo_epochs as a hyperparameter, setting it to 1 or 4. Could the authors explicitly clarify the definition of UTD and explain the precise difference between it and the ppo_epochs hyperparameter in their training setup?
> > >
> >
> > In the new version, we have provided a detailed explanation to improve clarity. In classic RL literature, UTD ratio refers to the number of gradient updates per batch of collected data. In our LLM RL infrastructure, the actual hyperparameter that controls UTD ratio corresponds to "PPO_epochs". Thus, **UTD and ppo_epochs are equivalent in our setup**, and we have updated the manuscript accordingly.
> >
> > ---
> >
> > Thanks  for the constructive feedback!  We believe these clarifications and additional experiments address the concerns raised and strengthen the contribution of the paper. Please let us know if further details would be helpful.

---

### Meta-Review · Area_Chair_zpLP · 2026-01-08

**Summary:**

This paper introduces Asymmetric Proximal Policy Optimization (AsyPPO), a framework that reinstates the critic's role in RL for LLM reasoning while maintaining computational efficiency. The key innovations include: (1) using lightweight "mini-critics" trained on disjoint prompt-level data shards to encourage diversity while preserving calibration, (2) leveraging inter-critic agreement to mask low-informative advantages, and (3) filtering high-divergence states from entropy regularization.

During the reviewing period, the main raised concerns including the scope of experimental validation, performance gains in some settings, and the need for more comprehensive baselines.

The idea of this paper is interesting. While the algorithm design is somewhat complicated and lacks theoretical grounding, the extensive rebuttal experiments have strengthened the empirical evidence and addressed most reviewer concerns.

**Reviewer Concerns:**

During the rebuttal period, the authors addressed most concerns through extensive additional experiments. Reviewer FX1U explicitly raised their score from 4 to 6. The concerns about theoretical grounding and the number of critics remain, but are not critical blockers given the empirical improvements.

**Reviewer Scores:**

Reviewer AgbJ likely remain 6, Reviewer FX1U 4 -> 6, Reviewer yexD 50% = 4, 50% -> 6

---

### Decision · Program_Chairs · 2026-01-26

Accept (Poster)